# Modulation of Glycinergic Neurotransmission may Contribute to the Analgesic Effects of Propacetamol

**DOI:** 10.3390/biom11040493

**Published:** 2021-03-25

**Authors:** Lukas Barsch, Robert Werdehausen, Andreas Leffler, Volker Eulenburg

**Affiliations:** 1Department of Anaesthesiology and Intensive Care, Medical Faculty, University of Leipzig, 04103 Leipzig, Germany; Lukas.Barsch@medizin.uni-leipzig.de (L.B.); Robert.Werdehausen@medizin.uni-leipzig.de (R.W.); 2Department of Anaesthesiology and Intensive Care Medicine, Hannover Medical School, 30625 Hannover, Germany; leffler.andreas@mh-hannover.de

**Keywords:** glycine, neurotransmitter, transporter, pain, neuropathy

## Abstract

Treating neuropathic pain remains challenging, and therefore new pharmacological strategies are urgently required. Here, the enhancement of glycinergic neurotransmission by either facilitating glycine receptors (GlyR) or inhibiting glycine transporter (GlyT) function to increase extracellular glycine concentration appears promising. Propacetamol is a *N*,*N*-diethylester of acetaminophen, a non-opioid analgesic used to treat mild pain conditions. In vivo, it is hydrolysed into *N*,*N*-diethylglycine (DEG) and acetaminophen. DEG has structural similarities to known alternative GlyT1 substrates. In this study, we analyzed possible effects of propacetamol, or its metabolite *N*,*N*-diethylglycine (DEG), on GlyRs or GlyTs function by using a two-electrode voltage clamp approach in *Xenopus laevis* oocytes. Our data demonstrate that, although propacetamol or acetaminophen had no effect on the function of the analysed glycine-responsive proteins, the propacetamol metabolite DEG acted as a low-affine substrate for both GlyT1 (EC50 > 7.6 mM) and GlyT2 (EC50 > 5.2 mM). It also acted as a mild positive allosteric modulator of GlyRα1 function at intermediate concentrations. Taken together, our data show that DEG influences both glycine transporter and receptor function, and therefore could facilitate glycinergic neurotransmission in a multimodal manner.

## 1. Introduction

Chronic pain is a major healthcare problem, occurring in about one in five adult Europeans and U.S. Americans. Nearly half of them receive inadequate pain management [1]. Here, especially, the treatment of neuropathic pain is still challenging [2], since commonly used painkillers and therapeutics are often not effective. Frequently used drugs for chronic neuropathic pain conditions include gabapentin and pregabalin, which target voltage-gated calcium channels, and antidepressants like duloxetine that inhibit the re-uptake of noradrenaline and serotonin. In many patients, these substances cause severe side effects that limit their benefit [3]; therefore, new pharmacological strategies are urgently required. For the treatment of chronic pain, the facilitation of inhibitory neurotransmission, which is predicted to reduce the excitability of neurons within the pain-processing micro-network in the dorsal horn of spinal cord, is considered a promising approach [4]. The targeting of GABAergic synapses has proven difficult due to their high abundance in higher brain areas. In contrast, the enhancement of glycinergic neurotransmission, which is restricted at large to caudal parts of the central nervous system, might result in fewer side effects. The glycine concentration in the central nervous system is regulated by the glycine transporters, GlyT1 and GlyT2 [5]. GlyT1 is expressed in astrocytes and a subpopulation of glutamatergic neurons and is essentially involved in the regulation of extracellular glycine levels at inhibitory glycinergic synapses [6,7]. Additionally, GlyT1 has also been shown to contribute to the regulation of *N*-Methyl-d-Aspartate (NMDA) receptors, which utilize glycine as a co-agonist [8]. In contrast, GlyT2 is exclusively expressed in glycinergic neurons and transports glycine unidirectionally into the presynapse to refill presynaptic vesicles [9]. Previous studies in animal models for chronic pain have demonstrated that inhibitors of the glycine transport activity of both transporters have antiallodynic and antihyperalgesic potential. Numerous GlyT1 inhibitors like ALX5407 or Bitopertin have been shown to ameliorate the facilitated pain response in animal models for chronic pain [10,11,12]. Interestingly, substrates of these transporters like sarcosine and the lidocaine metabolite *N*-ethyl glycine (NEG) also displayed antihyperalgesic and antiallodynic potential, most likely by inhibiting GlyT1-mediated glycine transport by a competitive mechanism [13,14]. But also GlyT2 inhibitors like ALX1393 [10], ORG25543 [15] and *N*-arachidonyl glycine [12,16] were suggested to have therapeutic potential for the treatment of neuropathic pain [17]. Similar to the inhibition of the glycine transporter (GlyT) activity, positive (allosteric) modulation of glycine receptor activity has been shown to ameliorate pain symptoms in some animal studies [4,18,19]. Despite these promising results, there are, at least to date, only very sparse clinical data available to demonstrate the clinical efficacy of any of these substances [20].

To foster a fast translation to clinical application, we searched for substances that might have potential effects on glycine sensitive proteins, but that are already used in clinics. We focused on substances that display, by themselves or by one of their metabolites, structural similarities to known GlyT1 substrates (i.e., glycine, sarcosine, or *N*-ethylglycine). Here, we identified propacetamol, a *N*,*N*-diethylglycine ester of acetaminophen that was developed initially to increase its solubility for a possible intravenous application and to reduce hepatotoxicity by prolonged release of acetaminophen [21,22]. After systemic application of propacetamol, it is cleaved by esterase to acetaminophen and N’N’-Diethyl-glycine (Figure 1). In this study we assessed the effect of *N*,*N*-diethylglycine (DEG) on both glycine transporter (GlyT) and glycine receptor (GlyR) function in a *Xenopus laevis* oocyte expression system by a two-electrode voltage clamp approach. We demonstrate that DEG, but not acetaminophen or propacetamol, functions as a low affinity alternative substrate for both GlyT1 and GlyT2. Moreover, DEG shows partial agonistic activity on homomeric GlyRα1, but not on GlyRα2 or GlyRα3. Taken together, our data raise the possibility that propacetamol, in addition to its well-characterized function as a COX inhibitor via acetaminophen, might influence glycine-dependent neurotransmission via a multimodal mechanism.

## 2. Materials and Methods

### 2.1. Drugs and Substances

All chemicals, if not stated otherwise, were purchased from Sigma Aldrich (Taufkirchen, Germany), Carl Roth (Karlsruhe, Germany) or AppliChem (Darmstadt, Germany). Propacetamol (Brand name: Acetamol inj.) was purchased from Standard Pharm & Chem, LTD. (Tainan City, Taiwan). Acetaminophen (Brand name: Paracetamol Kabi) was purchased from Fresenius Kabi Deutschland GmbH (Bad Homburg, Germany).

### 2.2. cRNA-Synthesis and Microinjection

The cRNA for the oocyte injection was synthesized by using the mMessage mMachine SP6 Transcription Kit™ (Thermo Fisher Scientific, Waltham, MA, USA) on linearized plasmid DNA containing cDNAs encoding for the respective proteins, following the manufacturer’s instructions. For linearization of the plasmids, restriction enzymes XbaI or HindIII (New England Biolabs, Ipswich, MA, USA) were used. Linearized plasmid DNAs were purified by phenol-chloroform extraction and ethanol precipitation. The structural integrity and concentration of linearized cDNA and synthesized cRNA samples were verified by agarose gel electrophoresis and photometric measurement. The cRNA samples were stored at −80 °C until use. The cRNA injection into defolliculated *Xenopus laevis* oocytes (Ecocyte Bioscience, Dortmund, Germany) was performed under optical control using a microinjector (Nanoject II, Drummond Scientific Company, Broomall, PA, USA) with mineral oil filled borosilicate glass capillaries (GB100F, Scientific Products GmbH, Kamenz, Germany), which had a tip diameter of approximately <50 µm. After injection, oocytes were stored at 15 °C in ND96 (in mM NaCl 96; KCl 2; CaCl_2_ 1; MgCl_2_ 1; HEPES 5; pH 7.4) + gentamycin (10 mg/L) until further use. Injection and expression conditions are summarized in Table 1.

### 2.3. Perfusion Regime

All solutions were freshly prepared in ND96 without (-) calcium (in mM NaCl 96; KCl 2; MgCl_2_ 1; HEPES 5; pH 7.4) and used for a maximum of 4 days. Solutions containing propacetamol were prepared immediately before the experiment and used for up to 8 h, to prevent hydrolysis of the ester. For experiments requiring hydrolysed propacetamol, hydrolysis was induced by adjustment of the pH to 9.5 and incubation of the solution for more than 10 h at room temperature, prior to the experiments. Afterwards, pH was readjusted to 7.4.

### 2.4. Electrophysiology

All two-electrode-voltage-clamp (TEVC) recordings were performed using a TurboTec-03X amplifier (npi elctronics, Tamm, Germany), and a custom-made recording chamber with the membrane potential set to −50 mV. Only oocytes showing a stable leak current <300 nA at the beginning of the experiment were used. Oocytes were superfused with ND96-Ca^2+^. Solutions containing substances at the indicated concentration for 20 s followed by a washout for at least 30 s or until stable baseline currents were observed. All measurements were performed at room temperature (21–23 °C). Current traces were evaluated using the software Cell Works and Cell Works Reader (npi electronics, Tamm, Germany). Excel (Microsoft, Redmond, WA, USA) and Graphpad Prism 7 (Graphpad Software, Inc., San Diego, CA, USA) were used for statistical analysis. Data are presented as mean ± SEM in dose-response curves or boxplots with whiskers from minimum to maximum with median and quartiles.

## 3. Results

### 3.1. Functional Characterization of GlyTs

To characterize the possible effects of Propacetamol and its metabolites on the function of glycine responsive proteins, the respective proteins were expressed in *Xenopus laevis* oocytes by cRNA injection and functionally tested by the analysis of current responses to superfusion with various glycine concentrations (Figure 2A). Whereas non-injected oocytes showed no current response to any of the substances used in this study, substance-induced currents were observed in recordings from GlyT1 and GlyT2-expressing oocytes. Here, already glycine concentrations of 1 µM glycine resulted in a robust current response. Maximal glycine-induced current amplitudes were observed at 1000 µM glycine with a mean amplitude of 109 ± 4.2 nA (n = 148), and an EC_50_ value of 24.9 µM (95%KI: 23.7–26.0 µM; n = 10–22) for GlyT1 and 128 ± 6.3 nA (n = 130) with an EC_50_ value of 13.2 µM (95%KI: 11.0–15.5 µM; n = 13–23) for GlyT2 (Figure 2B,C; Table 2).

### 3.2. GlyT Responses to Propacetamol and Paracetamol and DEG

To test if propacetamol or acetaminophen had any effect on the activity of GlyT1 or GlyT2, GlyT-expressing oocytes were superfused with each of these substances. Neither propacetamol nor acetaminophen alone produced any current response. Alkaline pretreatment, however, which is predicted to cause hydrolysis of the propacetamol (10 mM) to acetaminophen and DEG, resulted in significant inward currents in both GlyT1 and GlyT2-expressing oocytes, reaching 20–40% of the maximum current amplitude. Currents of slightly larger amplitudes (50–70%) were observed after the application of 10 mM DEG alone. Adding 20 µM glycine to 10 mM acetaminophen and 10 mM propacetamol solution each did not significantly alter the current response of any GlyT compared to 20 µM glycine solution alone (Figure 3A,B).

### 3.3. Characterization of the Metabolite DEG

To examine the function of DEG on GlyT function more precisely, its effects after application alone or after co-application with glycine were analysed. First, oocytes were superfused with DEG solution at increasing substance concentrations (100 µM, 333 µM, 1 mM, 3.3 mM, 10 mM), and glycine solution (1 mM). In addition, oocytes were superfused with sarcosine, which was shown previously to function as an alternative substrate for GlyT1 only, but not for GlyT2. Due to unstable measurements at higher DEG concentration, possibly caused by osmotic effects, only solutions of up to 10 mM DEG were applied to the oocytes. GlyT1 showed a current response during the application of glycine, sarcosine and also a dose-dependent current response to DEG at increasing concentrations. Sarcosine (1 mM) resulted in current amplitudes almost identical to those observed after glycine (1 mM), thus corroborating that sarcosine functions as an alternative substrate on GlyT1. After DEG superfusion, substance-induced currents were observed reliably at concentrations above 333 µM, reaching 57.6 ± 2.5% of the maximal glycine induced current at 10 mM DEG (Figure 4A).

Additionally, GlyT2 showed a dose-dependent current response to DEG. Sarcosine induced only insubstantial current responses, corroborating its specificity for GlyT1. DEG-induced currents were recorded reliably, starting from 333 µM, with maximal GlyT2-mediated currents observed at 10 mM, which corresponded to 67.7 ± 4.4% of the maximum glycine induced GlyT2 current (Figure 3a). Based on the dose–response curves obtained, EC_50_ values of >7.6 mM (lower 95% CI 7.0 mM, as determined by variable slope, best-fit values, least squares fit) for GlyT1 and >5.2 mM (lower 95% CI 4.6 mM) for GlyT2 were determined (Figure 4B, Table 3). These results demonstrate that DEG is a low affine substrate for both GlyTs, with a higher affinity for GlyT2.

Co-application of 20 µM glycine with 10 mM DEG showed a significant increase in GlyT1 (n = 13–23) activity as compared to 20 µM glycine, confirming DEG as a low-affine alternative substrate for GlyT1. Adding 3.3 mM DEG did not significantly influence GlyT1 activity at any glycine concentration. DEG did not influence GlyT2 (n = 15–31) in the presence of glycine at all investigated concentrations (Figure 4C).

### 3.4. DEG Influence on GlyR Activity

To determine if DEG influences GlyR function as well, its effects after application alone and after co-application with glycine on GlyR-expressing oocytes were analysed. In recordings from GlyRα1-3-expressing oocytes glycine induced currents were observed at concentrations starting from 33 µM reaching their maximal amplitude at 1000 µM (Figure 5A). A maximal amplitude of 6.1 ± 1 µA (n = 47) was observed in recordings from GlyRα1-expressing oocytes, 3.5 ± 0.4 µA (n = 45) in recordings from GlyRα2-expressing oocytes, and 2.4 µA ± 0.1 µA (n = 40) in recordings from GlyRα3-expressing oocytes (Figure 5C; Table 2). The dose–response curves obtained from these recordings revealed EC_50_ values for the respective receptor of 162 µM (95% CI: 141–186 µM; n = 12) for GlyRα1, 325 µM (95% CI: 289–362 µM; n = 3–15) for GlyRα2 and 234 µM (95% CI: 220–248 µM; n = 11) for GlyRα3 (Figure 5B; Table 2).

Although oocytes expressing any of the GlyR subunits showed great current response after application of 1 mM glycine, only GlyRα1-expressing oocytes showed a robust current response after sarcosine (1 mM) application. All GlyR-expressing oocytes showed only negligible current responses after perfusion with DEG (Figure 6a, Table 3). Upon co-application of DEG and glycine, however, the current response of GlyRα1-expressing oocytes to 20 µM glycine, which was 43.1 ± 3.3% of the maximum glycine-induced current amplitude (I_max_; n = 24–37), was increased to 58.1 ± 2.7% of I_max_ in the presence of 3.3 mM DEG (*p* ≤ 0.001). An increase in the DEG concentration to 10 mM, however, restored the initial state (47.7 ± 2.4% of I_max_). At higher glycine concentrations, a co-application of DEG resulted only in insignificant changes in the current amplitudes when compared to the current amplitudes elicited by glycine alone (Figure 6 B).

These findings suggest that DEG, in addition to its function at glycine transporters, is a low-affine partial agonist or allosteric modulator for GlyRα1. In recordings from GlyRα2 or GlyRα3-expressing oocytes, no effect of DEG application on glycine-induced currents was observed (Figure 6B; n = 14–30 and 12–29, respectively). Non-injected oocytes did not evoke any current response under substance application (Figure 2A; Figure 3A; Figure 4A), apart from unspecific osmolar effects at very high concentrations (data not shown).

## 4. Discussion

In this study, we show that the propacetamol metabolite DEG affects the function of GlyT1, GlyT2, as well as GlyRα1. In our experiments, we found that neither propacetamol nor acetaminophen showed any activity on the tested glycine-responsive proteins. After alkaline treatment of propacetamol, it was predicted to result in hydrolysis of propacetamol to acetaminophen and DEG; however, GlyT-specific current responses were observed. The recorded current amplitude reached 20–40% of the maximum current amplitude induced by glycine. Dose–response curves for DEG revealed EC_50_ values of above 5–8 mM for both GlyTs, supporting the hypothesis that the observed current response, observed in GlyT-expressing oocytes with alkaline-treated propacetamol, is caused by its hydrolysis product DEG. Our findings suggest that chemical hydrolysis probably led to a release of approximately 30–50% DEG. Enzymatic hydrolysis could further increase the amount of released DEG, and thereby enhance the effect of propacetamol in vivo.

Interestingly, DEG showed a substrate-like activity on both, GlyT1 and GlyT2. This contrasts with previous findings on mono-N-alkylated glycine derivatives that have been described previously as alternative substrates for GlyT1, but not for GlyT2 [23,24] (see also Figure 3). Similarly, the lidocaine metabolite *N*-ethylglycine (EG) was also shown to act as a substrate exclusively for GlyT1, with no additional activity on GlyT2 [25]. This specificity of mono-N’-alkylated glycine derivatives for GlyT1 was attributed previously to specific amino acid sidechains within the unwound region of TM6 [26], which was shown to form essential parts of the substrate-binding site in crystal structures of the bacterial GlyT ortholog LeuT_Aa_ [27], but also in other members of the Slc6A family like the *Drosophila* dopamine transporter DAT [28,29] and human serotonin transporter SERT [30]. Specifically, it was proposed that Ser-481 in GlyT2 is predicted to form a hydrogen bond to the amino group of a bound glycine substrate and poses a steric hinderance for substrates with substituents on the amino group. Consistently, substitution of this serine residue by glycine, which is present at the homologous position in GlyT1, also enabled sarcosine transport via GlyT2 [26]. Assuming binding of DEG to the primary substrate binding site S1 of the respective transporter, our current findings suggest that even substrates with large substituents on the amino group can be accommodated within the substrate binding site of both transporters. How the symmetrical ethyl groups bound to the amino group of DEG are accommodated in the substrate binding sites of both transporters, however, remains unclear at present.

During co-application experiments of glycine and DEG, a significant increase in the current amplitude with increasing DEG concentration was seen only in recordings from GlyT1, but not from GlyT2 expressing oocytes. Whereas for GlyT1, a synergistic activation of the transport associated currents by DEG and glycine was observed, as it is expected after application of substrates with different affinities for the transporter, a similar effect was not observed for GlyT2. The most likely explanation for this discrepancy is that GlyT2 shows a higher substrate selectivity for glycine than GlyT1. Whether this is caused by a different mode of DEG binding (which might be influenced by the presence of glycine) or by differences in the kinetics for the transport translocation (for glycine and DEG) is unclear at present.

Additionally, we demonstrate that DEG also influences the activity of GlyRα1, but not that of GlyRα2 or α3. In contrast to sarcosine, which showed agonistic activity on GlyRα1, but not at α2 or α3, as observed previously [13,31], DEG did not induce any significant current response by itself after perfusion of oocytes expressing any of the glycine receptors. These findings demonstrate that DEG does not function as a full agonist on the GlyRs. Interestingly, the glycine-induced GlyRα1 activity was significantly facilitated in the presence of intermediate DEG concentrations (3.3 mM), but not in the presence of higher concentrations. The structural similarities between glycine and DEG suggest that DEG most likely binds to the glycine binding site that is located within the N-terminal extracellular domain at the interface between two adjacent subunits [32]. Thus, DEG most likely functions as a partial agonist, although a function as an allosteric modulator with a binding site independent of the glycine binding site cannot be excluded. Here, DEG might trigger conformational changes within GlyRα1 that mimic some of the aspects of glycine binding, and thereby increase the probability of glycine binding at other glycine binding sites of the receptor without triggering the channel opening. At higher DEG concentrations, multiple DEG molecules might bind to the receptor, which prevents efficient receptor activation. This hypothesis is supported by the fact that only intermediate concentrations of DEG result in a facilitation of glycine response of GlyRα1-expressing oocytes, whereas this facilitation was not detectable at higher DEG concentrations.

Taken together, our data demonstrate that DEG is a low-affine substrate for both GlyT1 and GlyT2, and a partial agonist or allosteric modulator on the GlyRα1. It was recently demonstrated, via partial inhibition of GlyT2 and modest GlyR1 potentiation using a dual action compound, that substances moderately effective in modulating both GlyTs and GlyRs function may still be very effective in altering glycinergic neurotransmission [33]. This highlights the potential of multi-action substances like DEG. Although the DEG concentration needed to provoke a current response at GlyTs and GlyRα1 is high, it might be of some relevance in vivo, since it effects glycinergic neurotransmission via multiple targets. Here, the inhibition of GlyT1 and GlyT2, in addition to the partial agonistic activity on GlyRα1, homoreceptors might synergistically elicit antinociceptive effects and therefore might compensate its high dosage compared to other specifically GlyT modulating substances, like sarcosine [34] or EG [13,35]. The pharmacokinetic properties of propacetamol and its supposed enzymatic metabolization might, therefore, contribute to prolonged effects of propacetamol for chronic pain treatment. Our data suggest that propacetamol might have a previously unrecognized GlyRα1-modulating route of action by its metabolite DEG in addition to its known properties like cox inhibition. Since previously published data from patients with loss-of-function mutations of GlyRα1 indicate markedly reduced pain thresholds [36], propacetamol may prove effective in pain modalities that have previously been associated with diminished glycinergic neurotransmission like neuropathic pain. Moreover, DEG might also facilitate glycincergic neurotransmission mediated by other GlyR subtypes like GlyRα*3*, which has been implicated in the hyperalgesia and allodynia resulting from inflammation [37]; here, it might increase the local glycine concentration by its inhibition of glycine transporter via GlyT1 and GlyT2.

Previous studies on the efficacy of propacetamol showed a similar efficacy as compared to equimolar concentrations of acetaminophen [21]. These studies, however, focused exclusively on acute pain, such as post-surgery pain. Previous studies suggest only minor, if any, modulatory effects of glycinergic neurotransmission. Pathological pain, like neuropathic or long-lasting inflammatory pain, have not yet been –sed. Thus, future studies should investigate potential additional effects in animal models of chronic pain in order to evaluate the potential benefits for chronic pain patients.

## Figures and Tables

**Figure 1 biomolecules-11-00493-f001:**
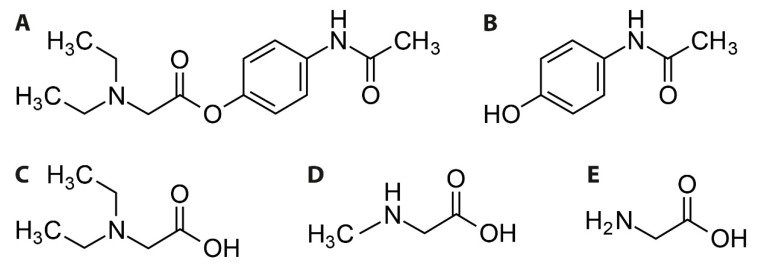
Chemical structure of (**A**) propacetamol, (**B**) acetaminophen, (**C**) *N*,*N*-diethylglycine (DEG), (**D**) sarcosine, (**E**) glycine. Propacetamol is hydrolysed to acetaminophen and DEG, which shares structural similarities to the neurotransmitter glycine.

**Figure 2 biomolecules-11-00493-f002:**
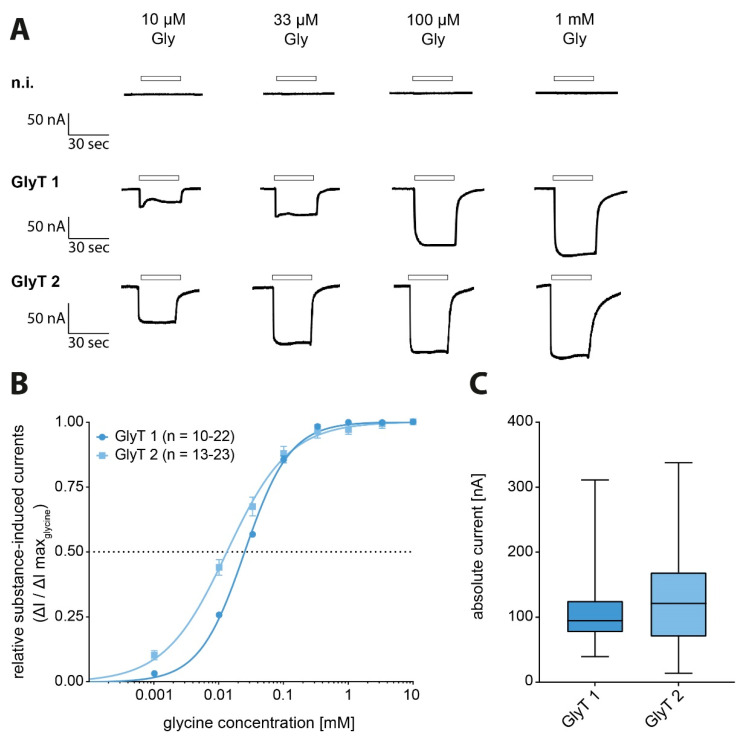
Functional analysis of glycine transporters (GlyTs) with glycine-induced dose-dependent current responses. (**A**) Representative current traces of original electrophysiological recordings from *Xenopus laevis* oocytes expressing GlyT1 or GlyT2, compared to non-injected oocytes after superfusion with glycine solution in increasing concentrations. Substance application is indicated by open box above the trace. (**B**) Dose–response curves of GlyTs showing the glycine-induced currents in relation to the maximum observed current induced by 10 mM glycine (GlyT1 n = 10–22, GlyT2 n = 13–23). (**C**) Maximum glycine-induced current amplitudes (nA) determined on oocytes expressing transporter individually taken together from all measurements (GlyT1 n = 148; GlyT2 n = 130; data presented as boxplots).

**Figure 3 biomolecules-11-00493-f003:**
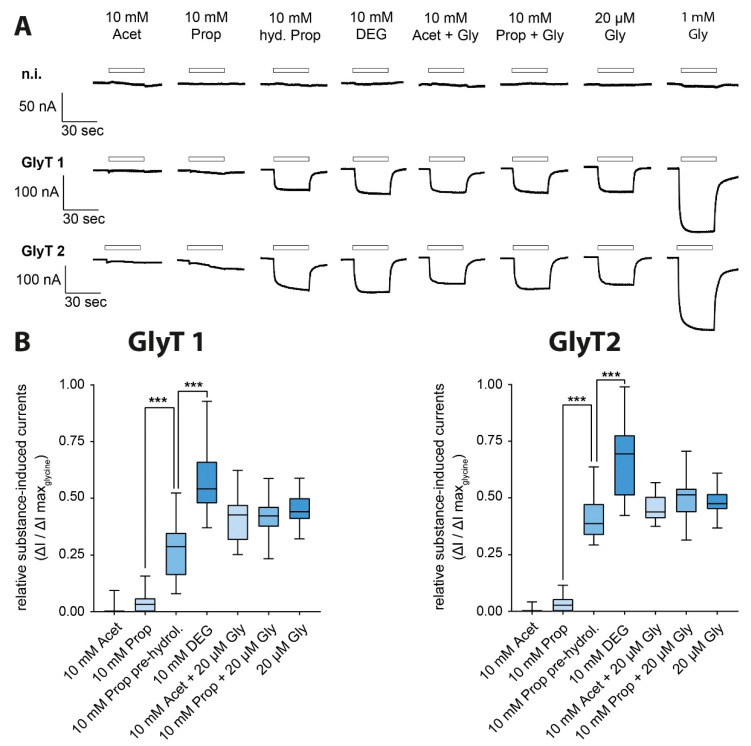
*N*,*N*-diethylglycine (DEG), but not propacetamol or acetaminophen, functions as an alternative substrate on GlyT1 and GlyT2. (**A**) Representative extract of original electrophysiological traces in *Xenopus laevis* oocytes expressing GlyT1 and GlyT2 individually compared to a non-injected oocyte after superfusion with glycine solution (20 µM, 1 mM), DEG (10 mM), acetaminophen (10 mM), propacetamol (10 mM), hydrolysed propacetamol (10 mM) and co-application with glycine (20 µM). (**B**) Relative substance-induced currents of GlyT1 and GlyT2 in relation to the maximum observed current induced by 1 mM glycine (GlyT1 n = 29–51; GlyT2 n = 13–44; data presented as boxplots, *p* < 0.001 (***), one-way ANOVA with Bonferroni post-hoc correction).

**Figure 4 biomolecules-11-00493-f004:**
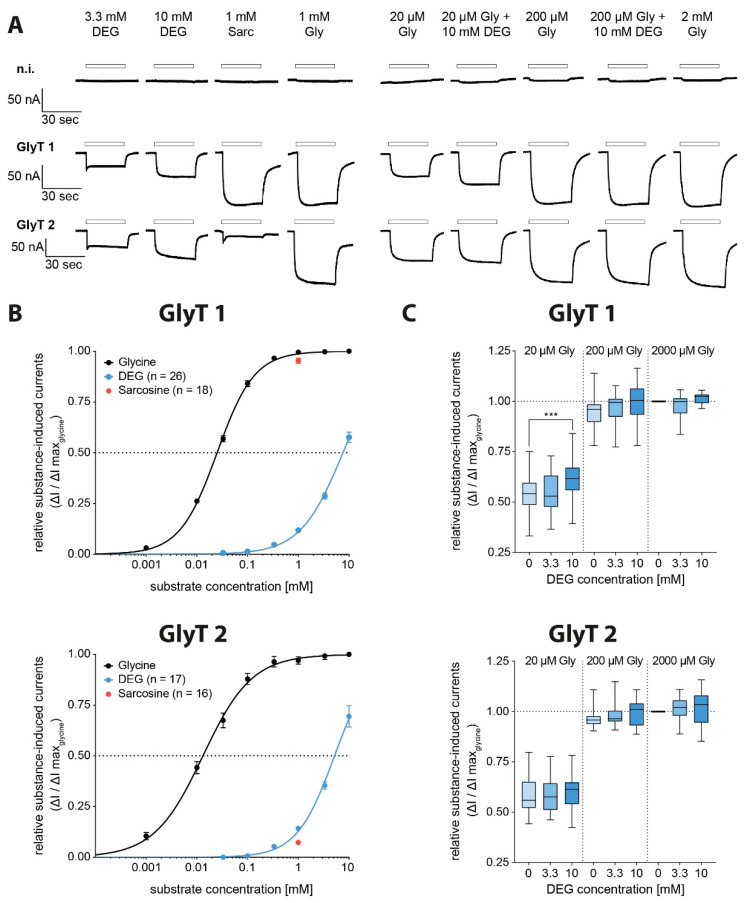
DEG functions as a full agonist at GlyT1 and GlyT2. (**A**) Representative original traces of electrophysiological measurements in *Xenopus laevis* oocytes expressing GlyT1 and GlyT2 individually after superfusion with DEG solution in increasing substance concentration (100 µM, 333 µM, 1 mM, 3.3 mM, 10 mM), sarcosine solution (1 mM) and glycine solution (1 mM) and after superfusion with different glycine solutions (20 µM, 200 µM, 2 mM) additionally containing distinct amounts of DEG (0 mM, 3.3 mM, 10 mM) to create co-application conditions. (**B**) Dose–response curves of GlyTs showing DEG and sarcosine-induced currents in relation to the maximum observed current induced by 1 mM glycine in comparison to the construct’s glycine dose–response curve (Figure 2B) (GlyT1: DEG n = 8–26, sarcosine n = 19; GlyT2: DEG n = 17, sarcosine n = 16). (**C**) Relative substance-induced currents of GlyT1 and GlyT2 in relation to the maximum observed current induced by 2 mM glycine + 0 mM DEG solution (GlyT1 n = 13–23; GlyT2 n = 15–31; data shown as boxplots, *p* < 0.001 (***), one-way ANOVA with Bonferroni post-hoc correction).

**Figure 5 biomolecules-11-00493-f005:**
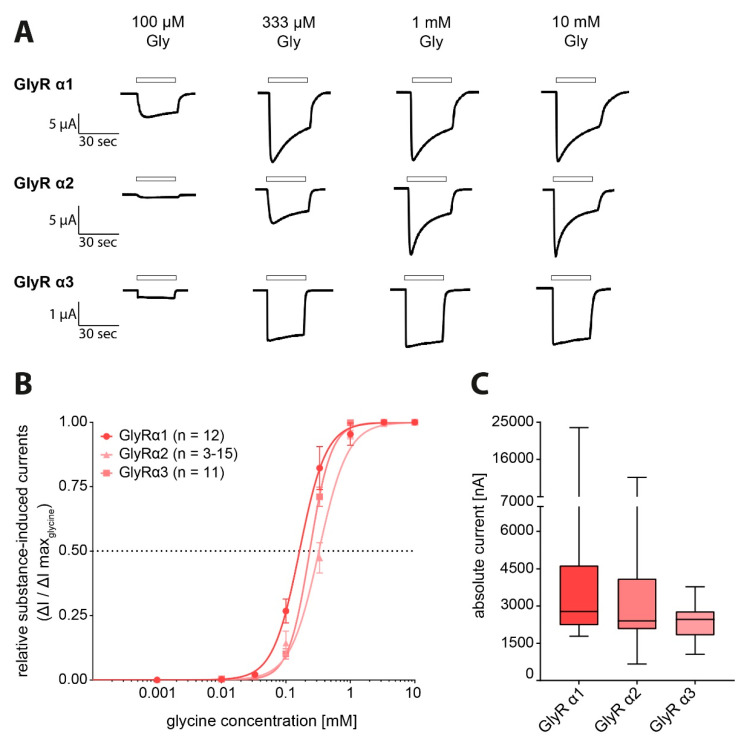
Functional analysis of the used glycine receptor (GlyR) constructs with glycine-induced dose-dependent current responses. (**A**) Representative extract of original electrophysiological traces in *Xenopus laevis* oocytes expressing GlyR subunits (α1-3) individually with glycine solution in increasing substance concentration (1 µM, 10 µM, 33 µM, 100 µM, 333 µM, 1 mM, 3.3 mM, 10 mM). Substance application is indicated by black bars. Note that due to limited perfusion channel number and differing substance concentrations in each experiment, not every measurement contained all concentrations. (**B**) Dose-response curves of GlyRs showing the glycine-induced currents in relation to the maximum observed current induced by 10 mM glycine (GlyRα1 n = 12, GlyRα2 n = 3–15, GlyRα3 n = 11). (**C**) Maximum glycine-induced current amplitudes [nA] determined on oocytes expressing transporter or receptor proteins individually taken together from all measurements (GlyRα1 n = 47, GlyRα2 n = 45, GlyRα3 n = 40; data shown as boxplots).

**Figure 6 biomolecules-11-00493-f006:**
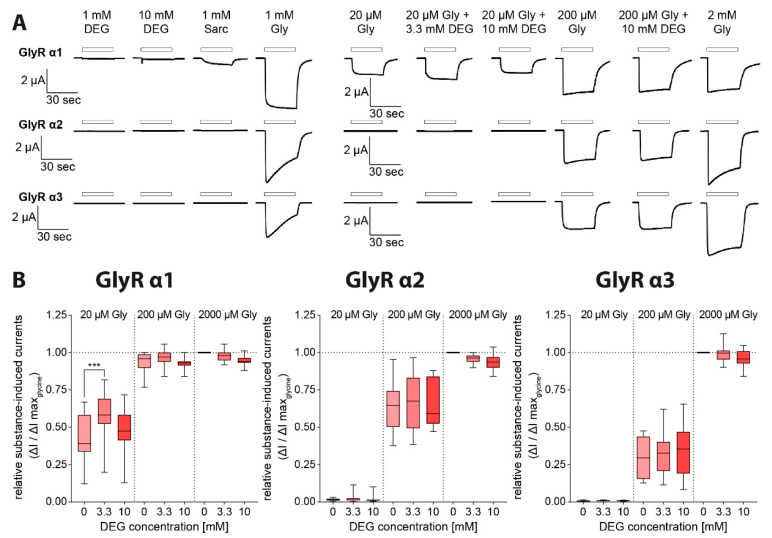
Current response of GlyRα1-3 after DEG application with and without glycine. (**A**) Original traces of electrophysiological measurements in *Xenopus laevis* oocytes expressing GlyRα1-3 individually after superfusion with DEG solution in increasing substance concentration (100 µM, 333 µM, 1 mM, 3.3 mM, 10 mM), sarcosine solution (1 mM) and glycine solution (1 mM) and after superfusion with different glycine solutions (20 µM, 200 µM, 2 mM) additionally containing amounts of DEG (0 mM, 3.3 mM, 10 mM). (**B**) Relative substance-induced currents of GlyRα1-3-expressing oocytes in relation to the maximum observed current induced by 2 mM glycine (GlyRα1 n = 24–37; GlyRα2 n = 14–30; GlyRα3 n = 12–29; data shown as boxplots, *p* < 0.001 (***), one-way ANOVA with Bonferroni post-hoc correction).

**Table 1 biomolecules-11-00493-t001:** cRNA synthesis and microinjection.

Construct	Plasmid	Restriction Enzyme	Concentration cRNA	Injection Volume	Incubation Time (d)
GlyT1	pNKS2(CG1) pNKS2(NG1)	XbaI	1 µg/µL	46 nL	4
GlyT2	pNKS2(CG2) pNKS2(NG2)	XbaI	1 µg/µL	46 nL	4
GlyRα1	pNKS2(GlyRα1)	XbaI	0.1 µg/µL	46 nL	1–2
GlyRα2	pNKS2(GlyRα2)	HindIII	0.1 µg/µL	46 nL	1–2
GlyRα3	pNKS2(GlyRα3)	XbaI	0.1 µg/µL	46 nL	1–2

**Table 2 biomolecules-11-00493-t002:** Electrophysiological data of glycine application.

Construct	Glycine Dose–Response Relationship			Maximal Glycine-Induced Current Amplitude	
	EC_50_ (µM)	95%CI (µM)	n	Mean ± SEM (nA)	n
GlyT1	24.9	23.7–26.0	10–22	109 ± 4.2	148
GlyT2	13.2	11.0–15.5	13–23	128 ± 6.3	130
GlyRα1	162	141–186	12	6075 ± 971	47
GlyRα2	325	289–362	3–15	3466 ± 387	45
GlyRα3	234	220–248	11	2359 ± 106	40

**Table 3 biomolecules-11-00493-t003:** Summary of electrophysiological data of DEG application.

Construct	DEG Dose–Response Relationship (Variable Slope, Best Fit Values, Robust Fit)			
	EC_50_	Lower 95%CI	Mean (I_DEG_/I_maxgly_) ± SEM	n
GlyT1	>7.6 mM	7.0 mM	57.6 ± 2.5%	8–26
GlyT2	>5.2 mM	4.6 mM	67.7 ± 4.4%	17

## Data Availability

The data presented in this study are available in the article.

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
