# Peer review of "Modulation of Glycinergic Neurotransmission may Contribute to the Analgesic Effects of Propacetamol"

_biomolecules, 2021, doi:10.3390/biom11040493_

Round 1
Reviewer 1 Report
In this work, Barsch et al. analyze the effects of N,N-diethylglycine (DEG), a metabolite of the non-opioid analgesic Propacetamol, on the glycine-responsive proteins the glycine receptor (GlyR) and the glycine transporters GlyT1 and GlyT2. By using two-electrode voltage clampin xenopus laevis oocytes, they found DEG exerts a low affinity inhibition of the transporters and a mild positive allosteric modulation of GlyR. The work is well designed and properly addressed and deserves publication. However, there are several points that require clarification before publication.
Major
- There is an apparent discrepancy. DEG has slightly higher affinity for GlyT2 but no effect in the presence of 20uM glycine whereas for GlyT1, which shows apparent lower affinity, DEG displays an effect in the presence of the same glycine concentration. How do the authors explain this apparent discrepancy? This has to be discussed.
- The effect of DEG on GlyRa1 displays a complex behavior increasing current amplitude only at intermediate concentrations. This behavior of DEG deserves a more robust possible explanation in the discussion (third paragraph).
- The authors state they notice unspecific osmolar effects at high concentrations of DEG. However, they do not notice these effects using 10 mM glycine. How do they explain these different effects?
- Discussion, second paragraph: The structural comments are not accurate. The binding site of DEG is not known at present. It is not clear that DEG binds to the S1 site. This sentence has to be rewritten.
- The role of GlyRa3 in pain transmission is relevant than that of GlyRa1. The authors have to discuss this in the context of the possible clinical action of Propacetamol.
Minor
- There are some typographic or grammar errors.
-Introduction: second paragraph should not be a full stop, but still first paragraph.
-Line 63 antiallydonic should say antiallodynic.
-Line 68 animal studies is not correct.
-Table 1: Fifth column “dilution factor” indicates the cRNA concentration for the GlyRs is 0.1 microgram/ml. The “dilution factor” column can be eliminated and the actual cRNA concentration corrected in the fourth column.
-Legend to figure 1, line 158: or receptor proteins should be removed.
-Line 165: first sentence of this paragraph is not correct “both substances” seem to indicate both are tested together and this is not the case. Please rewrite the sentence.
-Legend to figure 4: line 242 conbstructs should say constructs.
Author Response
First of all we would like to that the reviewer for his/her insightful comments that helped us to improve significantly the quality of the manuscript. A point to point reply is given below.
In this work, Barsch et al. analyze the effects of N,N-diethylglycine (DEG), a metabolite of the non-opioid analgesic Propacetamol, on the glycine-responsive proteins the glycine receptor (GlyR) and the glycine transporters GlyT1 and GlyT2. By using two-electrode voltage clamp in xenopus laevis oocytes, they found DEG exerts a low affinity inhibition of the transporters and a mild positive allosteric modulation of GlyR. The work is well designed and properly addressed and deserves publication. However, there are several points that require clarification before publication.
Major
There is an apparent discrepancy. DEG has slightly higher affinity for GlyT2 but no effect in the presence of 20uM glycine whereas for GlyT1, which shows apparent lower affinity, DEG displays an effect in the presence of the same glycine concentration. How do the authors explain this apparent discrepancy? This has to be discussed.
The reviewer is right. To discuss this discrepancy, we have extended the discussion at lines 308-317 by the following paragraph:
Interestingly, during coapplication experiments of glycine and DEG, a significant increase in the current amplitude with increasing DEG concentration was seen only in recordings from GlyT1, but not GlyT2 expressing oocytes. Whereas for GlyT1, a synergistic activation of the transport associated currents by DEG and glycine was observed, as it has to be expected after application of substrates with different affinities for the transporter, a similar effect was not observed for GlyT2. The most likely explanation for this discrepancy is that GlyT2 shows a higher substrate selectivity for glycine than GlyT1. Whether this is caused by a different mode of DEG binding (which might be influenced by the presence of glycine) or by differences in the kinetics for the transport translocation (for glycine and DEG) is unclear at present.
The effect of DEG on GlyRa1 displays a complex behavior increasing current amplitude only at intermediate concentrations. This behavior of DEG deserves a more robust possible explanation in the discussion (third paragraph).
We agree with the reviewer, that a more extensive discussion of this specific point is appropriate. We have rewritten the third paragraph of the discussion (line 318- to 336), which now states:
Additionally, we demonstrate that DEG also influences the activity of GlyRα1, but not that of GlyRα2 or α3. In contrast to sarcosine, that showed agonistic activity on GlyR α1, but not at α2 or α3 as observed previously [1,2], DEG did not induce any significant current response by itself after perfusion of oocytes expressing any of the glycine receptors. These findings demonstrate that DEG does not function as a full agonist on the GlyRs. Interestingly, the glycine-induced GlyR α1 activity was significantly facilitated in presence of intermediate DEG concentrations (3.3 mM) but not in the presence of higher concentrations. The structural similarities between glycine and DEG, suggest that DEG most likely binds to the glycine binding site that is located within the N-terminal extracellular domain at the interface between two adjacent subunits [3]. Thus DEG most likely functions as a partial agonist, although a function as an allosteric modulator with a binding site independent of the glycine binding site cannot be excluded. Here, DEG might trigger conformational changes within GlyRα1 that mimic some of the aspects of glycine binding and thereby increase the probability of glycine binding at other glycine binding sites of the receptor without triggering the channel opening. At higher DEG concentrations, multiple DEG molecules might bind to the receptor, which prevents efficient receptor activation. This hypothesis is supported by the fact that only intermediate concentrations of DEG result in a facilitation of glycine response of GlyRα1-expressing oocytes, whereas this facilitation was not detectable at higher DEG concentrations.
The authors state they notice unspecific osmolar effects at high concentrations of DEG. However, they do not notice these effects using 10 mM glycine. How do they explain these different effects?
This is apparently a misunderstanding. In our recordings both 10 mM DEG and 10 mM glycine were applied and after both stable current traces were obtained. Only after application of higher DEG concentrations unstable current responses that were interpreted as osmolar induced instability of the recordings were observed. Since Vmax was already reached at lower concentration, we did not try higher glycine concentrations and therefore a comparison of the effects of DEG and glycine at concentrations higher than 10 mM has not been made.
Discussion, second paragraph: The structural comments are not accurate. The binding site of DEG is not known at present. It is not clear that DEG binds to the S1 site. This sentence has to be rewritten.
Yes, the reviewer is right. Although it is highly likely that an apparently transportable substrate of the transporter is bound to the substrate binding site S1 as we suggested, our experiments do not prove this fact. We have therefore rewritten the respective sentence (line 302 and following in the discussion):
Since we found DEG to be a transportable substrate of the GlyT1 and GlyT2, we assume that it binds to the primary substrate binding site S1 of the respective transporter. Based on this assumption one may speculate that substrates even with large substituents at the amino group can be accommodated within the substrate binding site of both transporters.
The role of GlyRa3 in pain transmission is relevant than that of GlyRa1. The authors have to discuss this in the context of the possible clinical action of Propacetamol.
Here, we can only partially agree. As stated by the reviewer, an important function of GlyRa3 has been identified in the context of inflammatory pain, where a phosphorylation-dependent inhibition of the GlyRa3 activity has been described. For chronic pain conditions with a neuropathic component, however, the situation is more complex. Here, GlyRa3 does not seem to play a major role. Consistently, GlyRa3 KO mice show unchanged pain behaviour in animal models for neuropathic pain. In contrast, data from patients with loss-of-function mutations of GlyRa1 indicate markedly reduced pain thresholds [4]. Our data suggest that propacetamol might have a previously unrecognized GlyRa1 modulating route of action by its metabolite DEG in addition to its known properties like cox inhibition. It might therefore prove effective in pain modalities that have previously been associated with diminished glycinergic neurotransmission like neuropathic pain.
We inserted the following paragraph to the discussion section of the revised manuscript (Lines 351 to 360) in order to address this point:
Our data suggest that propacetamol might have a previously unrecognized GlyRa1 modulating route of action by its metabolite DEG in addition to its known properties like cox inhibition. Since previously published data from patients with loss-of-function mutations of GlyRa1 indicate markedly reduced pain thresholds [4] propacetamol may prove effective in pain modalities that have previously been associated with diminished glycinergic neurotransmission like neuropathic pain. Moreover DEG might also facilitate glycincergic neurotransmission mediated by other GlyR subtypes like GlyRa3 that has been implicated in the hyperalgesia and allodynia resulting from inflammation [5]. Here it might increase the local glycine concentration by its inhibition of glycine transporter via GlyT1 and GlyT2.
Minor
There are some typographic or grammar errors.
-Introduction: second paragraph should not be a full stop, but still first paragraph.
Corrected as requested
-Line 63 antiallydonic should say antiallodynic.
Corrected as requested
-Line 68 animal studies is not correct.
The reviewer is right, in the mentioned references only in vitro pharmacological data are discussed. We have therefore now included an additional reference (Acuna et al. 2016) that describes the mentioned animal studies (line 59 of the revised manuscript) .
-Table 1: Fifth column “dilution factor” indicates the cRNA concentration for the GlyRs is 0.1 microgram/ml. The “dilution factor” column can be eliminated and the actual cRNA concentration corrected in the fourth column.
The table has been modified according to the reviewer’s suggestions
-Legend to figure 1, line 158: or receptor proteins should be removed.
Corrected as suggested by the reviewer
-Line 165: first sentence of this paragraph is not correct “both substances” seem to indicate both are tested together and this is not the case. Please rewrite the sentence.
We have changed the sentence in line 160 ff to:
To test if propacetamol or acetaminophen have any effect on the activity of GlyT1 or GlyT2, GlyT-expressing oocytes were superfused with each of these substances. Neither propacetamol nor acetaminophen alone did produce any current response.
-Legend to figure 4: line 242 conbstructs should say constructs.
Corrected as suggested by the reviewer
Reviewer 2 Report
The manuscript entitled “Modulation of Glycinergic Neurotransmission may contribute 2 the analgesic effects of Propacetamol” by Barsch et al, aimed to investigate the potential role of propacetamol and its metabolite N,N-diethylglycine (DEG) in the treatment of acute neuropathic pain. For this purpose, the authors studied the effect of the cited molecules on GlyRs or GlyTs function.
The work is very interesting, and the results could be a start point for the effectiveness of DEG in chronic neuropathic pain.
I have only minor suggestion:
Some part of the manuscript is not clearly explained (line 81-87).
The structure of propacetamol, acetaminophen and DEG should be inserted also in the text to facilitate the readers comprehension.
Author Response
We would like to thank the reviewer for his/her positive assessment of our manuscript. A detailed point to point reply is given below.
The manuscript entitled “Modulation of Glycinergic Neurotransmission may contribute 2 the analgesic effects of Propacetamol” by Barsch et al, aimed to investigate the potential role of propacetamol and its metabolite N,N-diethylglycine (DEG) in the treatment of acute neuropathic pain. For this purpose, the authors studied the effect of the cited molecules on GlyRs or GlyTs function.
The work is very interesting, and the results could be a start point for the effectiveness of DEG in chronic neuropathic pain.
I have only minor suggestion:
Some part of the manuscript is not clearly explained (line 81-87).
To address this point we have changed this part of the introduction to:
We demonstrate, that DEG but not acetaminophen or propacetamol acts as a low-affinity alternative substrate for both GlyT1 and GlyT2. Moreover, DEG shows partial agonistic activity on homomeric GlyR α1 but not on GlyR α2 or GlyR α3. Taken together our data raises the possibility, that propacetamol, in addition to the well-characterized properties of acetaminophen, might influence glycine-dependent neurotransmission via a multimodal mechanism. (lines 72-77)
The structure of propacetamol, acetaminophen and DEG should be inserted also in the text to facilitate the readers comprehension.
As requested by the reviewer we have now included a new Figure 1 that depicts the structures of the respective molecules. For the sake of clarity, we have additionally included the structures of glycine and sarcosine. The numbers of all other figures have been changed accordingly.